

# Apsu: a wireless multichannel receiver system for surface-NMR groundwater investigations

Lichao Liu[1], Denys Grombacher[1], Esben Auken[1], and Jakob Juul Larsen[2]

[1]Hydrogeophysics Group, Department of Geoscience, Aarhus University, 8000 Aarhus C, Denmark
[2]Department of Engineering, Aarhus University, 8200 Aarhus N, Denmark

**Correspondence:** Lichao Liu (lichao@geo.au.dk)

**Abstract.** Surface nuclear magnetic resonance (surface-NMR) has the potential to be an important geophysical method for groundwater investigations, but the technique suffers from poor signal-to-noise ratio (SNR) and long measurement times. We present a new wireless, multichannel surface-NMR receiver system (called Apsu) designed to improve SNR, field deployability and minimize instrument dead time. It is a distributed wireless system consisting of a central unit and independently operated data acquisition boxes each with three channels that measure either the NMR signal or noise for reference noise cancellation. Communication between the central unit and the data acquisition boxes is done through long distance WiFi and recordings are retrieved in real time. The receiver system employs differential coils with low-noise pre-amplifiers and high-resolution wide dynamic range acquisition boards. Each channel contains multi-stage amplifiers, short settling-time filters and two 24-bit analog-to-digital converters in dual-gain mode sampling at 31.25 kHz. The system timing is controlled by GPS clock and sample jitter between channels is less than 12 ns. Separated transmitter/receiver coils and continuous acquisition allow NMR signals to be measured with zero instrument dead time. In processed data, analog and digital filters causes an effective dead time of 4 ms. Synchronization with an independently operated transmitter system is done with a current probe monitoring the NMR excitation pulses. The noise density measured in a shorted-input test is $1.8 \text{ nV}/\sqrt{\text{Hz}}$. We verify the accuracy of the receiver system with measurements of a magnetic dipole source and by comparing our NMR data with data obtained using an existing commercial instrument. The applicability of the system for reference noise cancellation is validated with field data.

# 1 Introduction

Sustainable groundwater extraction is critical to ensure continuous access to drinking water and irrigation water in countries all over the world. Geophysical methods can supply spatially dense and cost-effective hydrogeological information to guide groundwater extraction (e.g., Auken et al., 2003; Yaramanci et al., 2002). Surface-NMR is one such method featuring a direct sensitivity to aquifer properties such as water content and pore geometry (Lehmann-Horn et al., 2012; Knight et al., 2012; Behroozmand et al., 2015). By stimulating the unbalanced spins of hydrogen in groundwater with an excitation pulse (generated





using a surface wire-loop transmitter) in the background of the static Earth's magnetic field, the surface-NMR receiver can record a free induction decay (FID) signal released as the spins realign with the static magnetic field. The signal amplitude is proportional to the water content of the investigated sample volume and to the square of the static magnetic field $B_0$ (Weichman et al., 2000; Legchenko and Valla, 2002; Hertrich, 2008). The signal frequency, known as the Larmor frequency

$f_L$, is determined by the proton gyromagnetic ratio and $B_0$. The Earth's magnetic field ranges from 25 µT to 65 µT, resulting in signal amplitudes on the order of nanovolts and Larmor frequencies ranging from 1 kHz to 2.8 kHz. These small measured signals can easily be swamped by external noise sources, including powerline harmonics, spherics, electric fence spikes and other anthropogenic installations (e.g., Larsen et al., 2014; Costabel and Müller-Petke, 2014). Besides these noise sources, the inherent noise in the receiver can also dominate the NMR signal. The low SNR of measurements presents a major obstacle to

the application of surface-NMR.

     The first surface-NMR instruments used a single coil for both excitation and signal measurement (Legchenko and Valla, 2002). Later instruments introduced dedicated reference coils for simultaneous noise measurements used for reference noise cancellation to improve SNR (Radic, 2006; Walsh, 2008). State-of-the-art multichannel surface-NMR instruments are centralized systems, where the primary coil and reference coils are wired to a central controller. Although these systems have

significantly advanced surface-NMR, the current methodology also has a number of shortcomings. First, requiring a physical link (i.e. a wire) between the reference coil and the central controller limits the maximum separation between the primary coil and reference coils. If the reference coil is placed too close to the primary coil it may unintentionally detect the NMR signal, which would lead to cancellation of the NMR signal in the processed data (Larsen and Behroozmand, 2016). Second, the link cables are laborious to lay out in the field and increase setup and tear-down times, especially in scenarios where a large number

of coils are used as in surface-NMR tomography (Legchenko et al., 2011; Jiang et al., 2014). Third, a hardware-related instrument dead time is introduced when the same coil is used for transmitting the excitation pulse and receiving the NMR signal. We define the instrument dead time to be the portion of the signal that cannot be measured because of switching electromechanical relays. The instrument dead time prevents measurement of the initial high-amplitude part of the FID signals and reduces the SNR (Dlugosch et al., 2011; Walsh et al., 2011). These shortcomings can be addressed by employing a different surface-NMR

instrumentation where one transmitter (Tx) coil is used to emit the excitation pulse and small dedicated receiver (Rx) coils, wirelessly connected to a central unit, continuously record the NMR signal and reference noise.

     Wirelessly operated coils eliminate the need for cables to physically link the Rx coils to the central controller and grant increased flexibility. This is beneficial for field work, for example, if several soundings are being performed in the same survey area, the same placement of the reference coils can be employed and only the transmitter and primary coil must be moved

between soundings. In two and three-dimensional groundwater investigations, a receiver system with an array of wirelessly connected signal receiving coils distributed throughout the study area has great potential to provide significant reductions in field measurement times.

     We present a new multichannel surface-NMR receiver system, called Apsu, consisting of a central unit (ApsuMaster) with wireless connections to data acquisition boxes (ApsuRx). Each ApsuRx unit can be connected with up to three Rx coils

measuring signal or noise. The Rx coils are directly attached to pre-amplifiers and short cables pass the amplified signals to


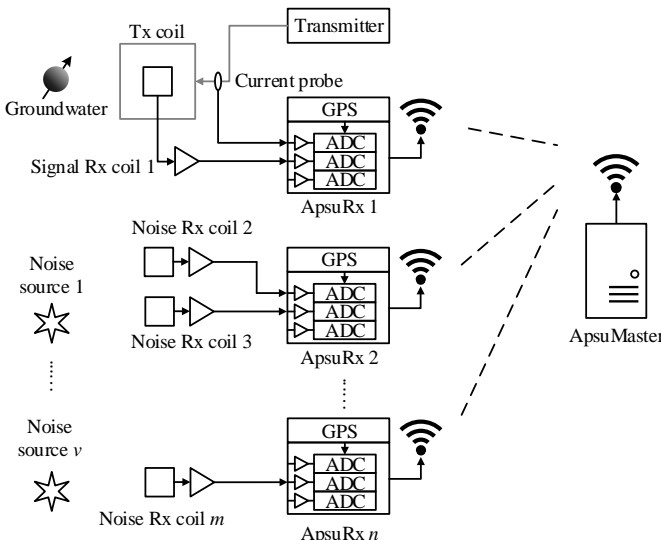

**Figure 1.** Setup of the wireless multichannel surface-NMR system Apsu. Each ApsuRx contains three GPS-synchronized channels. A current probe is connected to a channel of ApsuRx1 detecting the pulse, and signal Rx coil 1 is connected to the other channels recording NMR signal. In the vicinity of noise sources, ApsuRx2 - ApsuRx $n$ and noise Rx coil 2 - m are deployed.

the ApsuRx units where the signals are further processed using multi-stage amplifiers, bandpass filters and sampled by two 24-bit analog-to-digital converters (ADC) in dual-gain mode. Accurate timing is obtained by synchronizing the ApsuRx with the GPS clock. Communication and data transfer between the ApsuMaster and ApsuRx are done via long distance WiFi. The performance of the receiver system is validated with laboratory and field measurements, in particular a comparison with a Vista
Clara GMR system gives identical NMR signals.

## 2 Receiver design

The field setup of our wireless multichannel surface-NMR receiver system Apsu operating independently of a transmitter is shown in Fig. 1. One signal channel is dedicated to synchronizing the receiver system to an independently operating transmitter system by detecting the turn-on time and waveform of NMR excitation pulses using a current probe mounted on the transmitter
cable. One or more Rx coils are placed inside the Tx coil and connected to the primary channels for measuring NMR signal. Additional noise Rx coils and reference channels are placed distant to the Tx coil for continuous recording of noise. The channels used for signal detection and for noise recording are identical and they are configured remotely by the ApsuMaster. The ApsuMaster is a compact industrial PC running the instrument software and connected to an array of WiFi antennas. All channels are time-synchronized using GPS and are linked to the ApsuMaster through a WiFi network. The ApsuMaster retrieves
the measured time-series from all channels in real-time for processing and display. In the case of a lost WiFi connection, data can be retrieved asynchronously in the field or be accessed with a FTP client after the measurement.



## 2.1 Rx coil with a pre-amplifier

The voltage induced in a Rx coil is proportional to the area-turn product and the signal amplitude derived from it should be significantly higher than the receiver inherent noise. Surface-NMR signals are inherently weak but they can be magnified by high quality factor Rx coils tuned with external capacitors. However, tuned Rx coils have long transient response times, which

contributes to a long filter settling time and gives rise to near-exponential decaying oscillations in the frequency band of NMR signals if impulsive excitations (i.e. lighting and electric fence discharge) are present. Moreover, the quality factors must be calibrated before measurements. To reduce filter settling time and calibration procedures, untuned Rx coils are used in our instrument.

In standard surface-NMR measurements, the area-turn product of Rx coils range from 1000 $\mathrm{m}^2$ to 10000 $\mathrm{m}^2$. Larger Rx

coils gives higher signal amplitude, but they take more time and manpower to deploy in surveys. Another drawback is that larger Rx coils have high inductance and parasitic capacitance, resulting in lower cutoff frequency and significant phase shift of the surface-NMR signal. In addition, SNR remains unimproved for larger coils, once the area-turn product is high enough. In this limit, the inherent noise in the receiver electronics is negligible compared to the signal and noise coupled inductively into the coil. Increasing the area-turn product gives a corresponding increase of both signal and coupled noise and SNR is

unimproved.

Based on noise measurements conducted at different sites in Denmark, a typical value for the stochastic background noise density is $0.02$ $\mathrm{nV}/(\mathrm{m}^2\sqrt{\mathrm{Hz}})$ (Nyboe and Sørensen, 2012). Considering an area-turn product of 1000 $\mathrm{m}^2$, the external noise will be about ten times stronger than our receiver inherent noise, as discussed later. The Rx coils used for the measurements presented here are 10 m × 10 m, eight-turn differential coils, Fig. 2, with resistance $R$=17.4 $\Omega$, self-inductance $L$=1.85 mH

and parasitic capacitance $C$=7.4 nF. The cutoff frequency of these Rx coils is 43 kHz, which is approximately 20 times higher than the local Larmor frequency, hence the amplitude and phase distortion of the NMR signal is negligible.

Damping resistors $R_d$ are used to set the Rx coil into the critically damped state or slightly over-damped. The damping resistance is given by (Lehtonen and Hällström, 2017),

$$R_d \geq \frac{L}{RC + 2\sqrt{LC}}. \tag{1}$$

A pre-amplifier is mounted directly at the coil to amplify the signal before transmission. The pre-amplifier is a very low-noise, low-distortion operational amplifier with a typical input noise voltage of 0.9 $\mathrm{nV}/\sqrt{\mathrm{Hz}}$ in the 1-3 kHz range. The amplifier gain is 21, determined by gain resistor $R_g$ and feedback resistor $R_f$. The low resistance resistors used in the pre-amplifier reduce the internal noise in the pre-amplifier.

The Rx coil and pre-amplifier are physically connected during the entire measurement, so the induced voltage can exceed the

maximum input range of the pre-amplifier during the surface-NMR transmit pulse. To protect against over-voltage, a clamper consisting of two diodes in inverse parallel is used to clip the voltage to a safe range if the amplitude exceeds the forward voltage of the diode. A current limiting resistor in series with the diodes eliminates the effect on the excitation magnetic field by the induced current in the Rx loop. The output signal from the pre-amplifier is sent to the acquisition board through a 5 m long twisted cable.





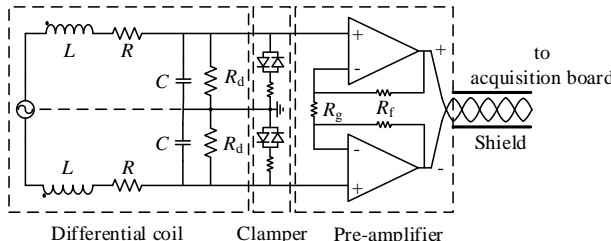

**Figure 2.** Schematic diagram of the differential Rx coil and pre-amplifier. A diode clamper is used to protect the amplifiers from over-voltage during strong Tx pulses.

The amplifiers used in the acquisition board have the same noise levels as the pre-amplifier, so the receiver inherent input noise density, $n_i$, is determined by the pre-amplifier noise. The noise density in the pre-amplifier is composed of thermal, voltage $e_n$ and current noise $i_n$,

$$n_i = \sqrt{2[4kT(R_s + R_a) + e_n^2 + (i_n R_a)^2 + (i_n Z)^2]}, \tag{2}$$

where $k$ is Boltzmanns constant and $T$ is the temperature and

$$R_s = \frac{R R_d}{R + R_d}, R_a = \frac{R_f R_g/2}{R_f + R_g/2},$$

$$Z = \frac{R_d(j\omega L + R)}{(j\omega L + R)(j\omega R_d C + 1) + R_d}.$$

The theoretical inherent noise density is 1.8 nV/$\sqrt{\text{Hz}}$ at 2 kHz. In measurements, an $N$-fold stacking reduces the inherent noise by a factor of $\sqrt{N}$.

## 2.2 Acquisition board

The acquisition board is a custom made board containing a low power consumption field-programmable gate array (FPGA) and three high resolution channels with multi-stage amplifiers, filters and 24-bit ADCs. A block diagram of the acquisition board is shown in Fig. 3.

The analog circuit of the acquisition board employs differential signaling in accordance with the Rx coil design to cancel common mode noise. The input amplifier is low-noise, with a fixed gain of 21. There are two stages of gain-controlled amplifiers (GCA) and their gain factors are remotely adjusted by the ApsuMaster according to the local noise level.

Although the Rx coils attenuate the noise with frequency higher than the cutoff frequency, the signals input to the ApsuRx are normally severely affected by externally coupled noise. Saturation of the second-stage GCA due to externally coupled noise is avoided by applying filter that remove out-of-band noise before the signal reaches this amplifier. The transient response of



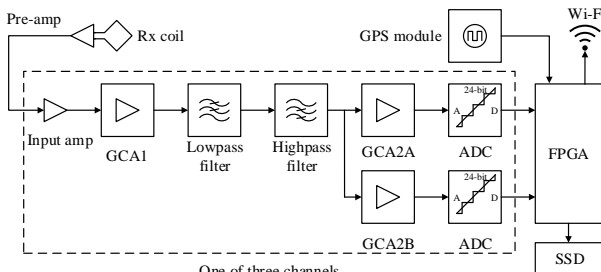

**Figure 3.** Block diagram of acquisition board. Note that each ApsuRx contains three duplicated channels, and only one of them is illustrated in the dashed box.

the bandpass filters contributes to the filter settling time. The transient response rate of a second-order filter is characterized by the settling-time $t_s$ (Angeles, 2011),

$$t_s = \frac{4}{\zeta \omega_c}, \qquad (3)$$

where $\zeta$ is the damping ratio and $\omega_c$ is the cut-off frequency. We utilize a second-order low-pass filter and a second-order
high-pass filter in cascade for noise reduction. The lowpass filter with a cut-off frequency of 5 kHz reduces radio frequency noise and the high-pass filter with a cutoff frequency of 1 kHz is responsible for reducing low-order powerline harmonics. The filters have an overall settling-time of only 0.42 ms preserving the early-time signal. The phase shift imposed on the NMR signal by the filters is determined by $f_L$ and can be modeled and calibrated.

The second GCA operates in dual-gain mode and the two signals are independently quantized by two 24-bit ADCs. The high
accuracy ADC has a typical SNR of 113.5 dB. In normal cases, the signal from the high-gain path is used to maintain a high resolution. In segments of data where strong noise, e.g. from spikes, saturates the high-gain path, the signal in the low-gain path can be substituted and unusable segments of data are avoided. To replace the saturated data in the high-gain path, all the samples are converted to the true input voltage after being divided by the gain factor. The dynamic range is increased by the gain factor ratio between the high-gain path and low-gain path.

To minimize the inherent noise of ApsuRx and obtain the optimal resolution, the overall gain factor through the measurement channel must be chosen so that the ADC quantization noise is much smaller than the noise from the analog circuit. The ADC quantization noise density, $n_{ADC}$, depends on the SNR of the ADC, $SNR_{ADC}$, and the sampling frequency $f_s$ (Walden, 1999),

$$n_{ADC} = \frac{V_{FS}}{10^{SNR_{ADC}/20}} \frac{1}{\sqrt{f_s/2}}, \qquad (4)$$

where $V_{FS}$ is the 5 V full scale voltage of the ADC. For a sampling frequency of 31.25 kHz, the ADC quantization noise density is 84.5 nV/$\sqrt{\text{Hz}}$. Considering the 1.8 nV/$\sqrt{\text{Hz}}$ analog circuit noise density, the overall gain factor must be at least 500 for a 10:1 ratio. The minimum and maximum gain factors of the ApsuRx amplifiers chain are 441 and 194481, respectively.





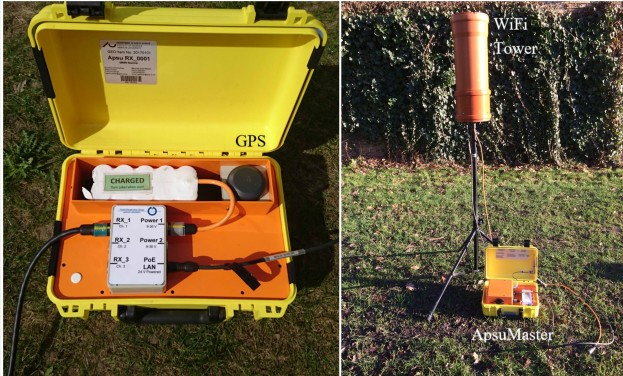

**Figure 4.** Left: ApsuRx. The break-out-box of ApsuRx provides plugs for three signal inputs, power and WiFi antenna connection. The GPS antenna in the top-right corner is used for time synchronization. Right: ApsuMaster is connected to the WiFi tower which consists of a array of WiFi antennas working in the AP mode.

The FPGA has a dual-core ARM processing system (PS) part and a programmable logic (PL) part in a single device. The acquisition board contains two I/O headers that connect to I/O banks on the PL side and is used as an embedded system-on-module. To fulfill the critical timing requirements in signal acquisition, blocks for synchronizing the wirelessly connected ApsuRx units and reading the ADCs are implemented in the PL part. Synchronization block is based on a D flip-flop triggered

by the rising edge of the GPS pulse and is remotely controlled by the ApsuMaster. The ADC reading block for each channel uses the serial peripheral interface (SPI) bus to read out the two ADCs in daisy-chain, which ensures the high-gain path and low-gain path are sampled simultaneously. Another block receives and decodes the time and location information from the GPS module. The GPS time is refreshed every second and used as the reference for timing synchronization between ApsuRx units.

The two ARM processors in the PS are programmed in asymmetric multi-processing (AMP) mode: one is running an embedded Linux system and the other executes the bare-metal applications. The AMP mode takes advantage of the efficiency of a bare-metal machine in real-time applications and the flexibility of the Linux system in networking and massive data operating programming. The peripherals of the PS include 1 GB of memory, a gigabit Ethernet, a micro SD and a USB port. The gigabit Ethernet is connected to a WiFi antenna and the USB port is connected to a 128 GB solid-state drive (SSD) for

data logging. The Linux system is responsible for communicating with the ApsuMaster through WiFi, storing and retrieving data, as well as controlling the bare-metal processor. The bare-metal processor is interrupt-driven and reads samples from the ADC reading block in the PL part and writes them into the shared memory. Subsequently, the data is grabbed and stored into the SSD by the Linux system. When the ApsuMaster requests data from the ApsuRx, the Linux system reads the data from the SSD and transfers it to the ApsuMaster through WiFi. ApsuRx also contains a FTP server and the collected data can be

accessed through a FTP client software after the measurement. The acquisition board, GPS antenna, and a rechargeable 9 Ah lithium battery are assembled in a waterproof suitcase, Fig. 4.

## 2.3 Synchronization between channels

The timing accuracy and stability of the receiver system are of fundamental importance for obtaining high quality data. Any significant timing jitter or uneven sampling frequencies between channels will significantly reduce the efficiency of the reference noise cancellation and phase determination of the NMR signal. Therefore, the system is designed with strict requirements

on timing. We utilize a GPS timing module to maintain high timing accuracy. The GPS module provides two outputs: a pulse per second (PPS) signal and a reference clock. The PPS pulse with a jitter $jt_{pps}$ less than 11 ns is used to trigger the sample-start clock of the ADCs. Triggering the sample-start clocks between the ApsuRx units is achieved with the following procedures: 1) the ApsuMaster sends a command to the ApsuRx units indicating the scheduled synchronization time; 2) all the ApsuRx units enable the synchronization blocks; 3) reset pulses for all the ADCs are generated at the rising edge of the scheduled PPS;

and 4) the channels output valid data after a fixed settling-time of the ADC's digital filters, which is 2370 cycles of the driving clock. Following these procedures, all ApsuRx units collect data continuously until the next synchronization action.

To ensure that all ApsuRx units use the identical sampling frequency, the ADCs are driven by a low-jitter 1 MHz GPS derived reference clock. The mean error $\mu(\Delta f_{clk})$ and standard deviation $\sigma(\Delta f_{clk})$ of the reference clock are $6.2 \times 10^{-5}$ Hz and $7.0 \times 10^{-4}$ Hz, respectively. The frequency of the reference clock and the decimation ratio of the ADC $\lambda = 32$ result in a

sampling rate $f_s$ of 31.25 kHz.

The jitter of the PPS, the instability of the reference clock and the lack of synchronization with the 200 MHz main clock $f_{mclk}$ of the FPGA all contribute to the sample-start jitter $jt_{sync}$ between channels, which is calculated as,

$$jt_{sync} = \sqrt{jt_{pps}^2 + (2370 \times \frac{\sigma(\Delta f_{clk})}{f_{clk}^2})^2 + (\frac{1}{f_{mclk}})^2}. \tag{5}$$

The GPS module only needs single satellite to provide accurate pulse timing and it works nicely in practice, even with lim-

ited field of view and cloud cover. Compared with the PPS jitter, the frequency error of $f_{clk}$ contributes very little to the synchronization jitter. The synchronization jitter $jt_{sync}$ is less than 12 ns which is much smaller than the 32 μs sample period.

Once the $m$th ADC is synchronized at time $t_{sync}(m)$, every new sample will be labeled with a unique time-stamp $t_{ID}(m,n)$. The recordings are stored in a data structure in the SSD. Each structure consists of a one-second time-series and the corresponding data header. The data header indicates the properties of the time-series, such as gain factors, synchronization time $t_{sync}(m)$

and time-stamps $t_{ID}(m,n)$. To align the samples from different channels after they are transfered to the ApsuMaster, the global time $t_g(m,n)$ should be equal for each sample,

$$t_g(m,n) = t_{sync}(m) + \frac{t_{ID}(m,n)}{f_s}. \tag{6}$$

Due to the frequency error of the ADC clock in the different channels, there is a time shift $t_{shift}$ between the two aligned samples after the channels have been synchronized for $T_N$ seconds. The time shift between samples is written as,

$$t_{shift} = jt_{sync} + \lambda T_N f_s \frac{\mu(\Delta f_{clk})}{f_{clk}^2}. \tag{7}$$

A time shift of less than 0.22 μs is generated per hour in our multichannel receiver system. Hence, the time shift in one measurement can be ignored compared to the cycle of the NMR signal.



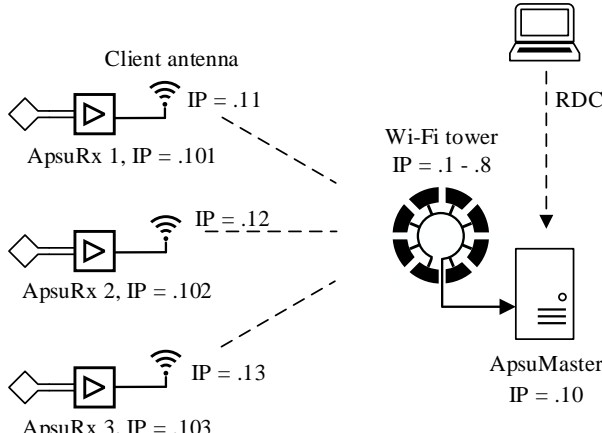

**Figure 5.** Topology of the network of wireless surface-NMR instrument Apsu. The network mask is 255.255.255.0 and only the host numbers of the IP address are shown.

## 3 Wireless communication network

A design goal of the system is to be able to conduct surface-NMR measurements with wireless communication to remote coils located up to 1 km away. Besides coverage, link speed is another crucial requirement of the network. A 1 s time-series sampled

in 24 bits at 31.25 kHz plus data header amounts to 96 kB and as each ApsuRx unit contains three channels the transfer speed should be higher than 288 kB/s for real time operation. We used an outdoor long distance WiFi, which has a dual-polarized directional antenna and a high transmit power. This WiFi antenna provides a link speed up to 300 Mbps and a maximum coverage range of 10 km along the pointed direction.

### 3.1 WiFi network

The wireless network consists of the ApsuMaster and a number of ApsuRx units, Fig. 5. The ApsuMaster is based on a high performance, compact industrial PC installed in a solid, waterproof case, Fig. 4. A WiFi tower consisting of an array of WiFi antennas working in access point (AP) mode is connected to the ApsuMaster. The measurements retrieved by the ApsuMaster can be monitored through a wireless remote desktop connection (RDC). The ApsuRx units are each attached to a single WiFi antenna set to client mode.

The signal beam width of each WiFi antenna is 45 ° in the horizontal plane and 30 ° in the vertical plane. To achieve full 360 ° coverage, eight WiFi antennas are used in the WiFi tower. The antennas are mounted at the top of an adjustable tripod and encapsulated in a waterproof container, Fig. 4. Each antenna provides power and digital link to the next WiFi antenna in the WiFi tower. By daisy-chaining the eight WiFi antennas, the WiFi tower can be connected to the ApsuMaster through a single Cat5e cable. The eight WiFi antennas connected to the ApsuMaster not only provide better coverage but also improve the overall network-capacity and transfer speed. The client WiFi antennas in the ApsuRx automatically connect to the AP in



WiFi tower which provides the strongest radio signal for it. To obtain stable wireless link, the horizontal and vertical angles of the WiFi antennas in the ApsuRx are adjusted to face the WiFi tower.

## 3.2 Communication protocol

Communication between the ApsuMaster and ApsuRx in the receiver system is done using stream sockets on top of TCP/IP. A server/client pair is established between the ApsuMaster and each ApsuRx. Each server/client pair runs in its own thread to keep communication with each ApsuRx unit independent. At start-up, each server socket in the ApsuMaster listens for a connection request from the corresponding ApsuRx. Once the connection is established, the client socket can send commands to the ApsuRx. In the ApsuRx, the server socket receives commands from the ApsuMaster and the client socket responds to it

by sending a data package to the ApsuMaster server socket. The communication between the ApsuMaster and each ApsuRx is a full-duplex transmission pipe. Software in the ApsuMaster handles the TCP/IP communication and processes the retrieved data.

A custom communication frame protocol is defined for the decoding of different commands from the ApsuMaster. The command frame sent from the ApsuMaster to the ApsuRx is a binary sequence including start code, frame length, command

ID and command parameters. By defining different command IDs both in the ApsuMaster and the ApsuRx, each ApsuRx can perform independent actions such as gain factor setting, channel synchronization, status retrieval and data transfer.

## 4  Performance verification

### 4.1  Shorted-input noise test

We verified the inherent noise properties of the receiver system by shorting the input to each pre-amplifier on two measurement

channels and recording noise simultaneously from the two channels. After subtraction of DC offsets, the samples recorded in one second are shown in Fig. 6(a).

The voltages are in the range of $\pm 500$ nV, and the root mean square (RMS) values $n_{rms}$ for the two channels are both 114 nV. We found no correlation between measurements from the two channels and the histogram distribution of voltages matches the probability density function (PDF) of Gaussian noise (Fig. 6(b)), implying that the measured voltages are receiver

inherent noise. The inherent noise density of the ApsuRx can be calculated as,

$$n_d = \frac{n_{rms}}{\sqrt{f_{high} - f_{low}}},\tag{8}$$

where $f_{low}$ and $f_{high}$ are the cut-off frequencies of bandpass filter in ApsuRx. The noise spectral density in a 1 second measurement is $1.8\pm0.3$ nV/$\sqrt{\text{Hz}}$ in the passband at 2 kHz. This measurement is in good agreement with the theoretical noise density of 1.8 nV/$\sqrt{\text{Hz}}$ in Sect. 2.1.




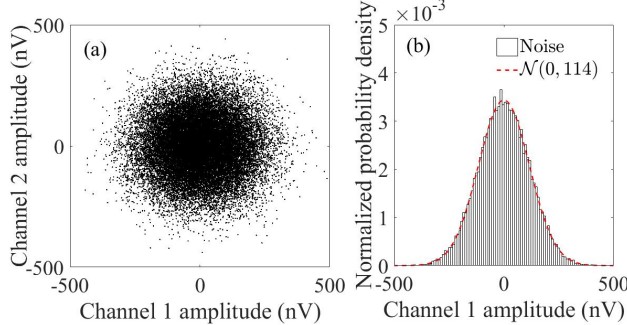

**Figure 6.** (a) Scatter plot of one second of samples from two channels recorded with shorted inputs on the pre-amplifiers, DC offsets are removed. (b) Normalized PDF histogram of channel 1 noise amplitudes. The width of the bins is 10 nV and there are 100 bins in total. The red dashed line is the PDF curve of the Gaussian distribution with zero mean and standard deviation of 114 nV.

## 4.2 Measurement accuracy

The accuracy of the receiver system was validated with measurements of a magnetic dipole source. The magnetic dipole was a one-turn, 5 m by 5 m square coil driven by a signal generator and an audio amplifier. The Rx coil was a 16-turn, 5 m by 5 m square coil. The center-to-center distance between the two coils was 50 m. The distance between the two coils was 10 times the side length and the transmitter magnetic field can therefore be approximated as a magnetic dipole. The theoretical receiver signal caused by the current in a coplanar transmitter coil is given as,

$$V(t) = \frac{\mu_0 \omega I(t) A_t A_r}{4\pi r^3}, \tag{9}$$

where $\mu_0$ denotes magnetic permeability in the air, $r$ is the distance between the coils, $A_t$ and $A_r$ are the area-turn products of the two coils. The signal generator output was a 2125 Hz sinusoid and the transmitter current was adjusted from 0.1 A to 5 A. The current $I(t)$ was determined by a commercial current probe attached to an oscilloscope. The receiver signal amplitudes were obtained by a Fourier transform of the measured time-series. The receiver signal amplitude is plotted against the transmitter current along with the theoretical prediction in Fig. 7. We observed that the received signal amplitude increased linearly with the transmitter current and all measurements match the theoretical values with a less than 1 % relative error, confirming the accuracy of the receiver system.

## 4.3 Reference noise cancellation

We tested the applicability of reference noise cancellation with wirelessly connected receiver coils using a synthetic surface-NMR signal embedded in noise-only records. The synthetic signal was chosen as this makes it possible to compare the output of the processed data with the expected result. Noise-only data were collected near Silkeborg, Denmark. One ApsuRx served as the 'signal' receiver and was connected to a Rx coil. A second ApsuRx, located approximately 200 m away, served as the





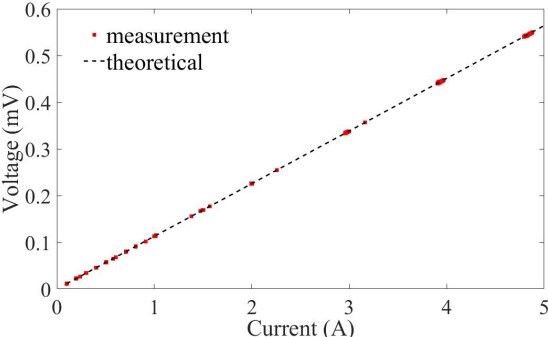

**Figure 7.** Comparison between measured (red squares) and theoretical (black line) signal amplitudes from a magnetic dipole source.

remote reference receiver and was connected to two Rx coils. All coils were 5 m by 5 m, 16-turn coils. The distance between the two reference coils was approximately 100 m.

The data from the three coils were dominated by powerline harmonics and stochastic background noise. A mono-exponential
synthetic NMR signal with parameters $E_0 = 100$ nV, $T_2^* = 200$ ms and $f_L = 2124$ Hz was embedded in the 'signal' data. Subsequently, the data was bandpass filtered with a 500 Hz bandwidth filter centered at the Larmor frequency. Powerline harmonics were subtracted from all data sets using the model-based method (Larsen et al., 2014). The RMS value of noise in the original data was 3320 nV. The RMS value decreased to 147 nV after bandpass filtering and to 85 nV after harmonic subtraction. Reference noise cancellation (RNC) using Wiener filtering and noise from two reference coils further reduced the
RMS value to 62 nV. Both the data before and after RNC are used to extract envelopes through digital quadrature detection and stacking. Figure 8 shows the two envelopes after quadratic detection and 18-time stacking, the blue solid line and black dashed line are the results processed without and with RNC.

Compared with the envelope without RNC, the result processed with RNC matches better with the synthetic envelope (red dotted line). The SNR for the two envelopes is 0.4 dB and 5.1 dB, respectively, which implies an SNR improvement of 4.7 dB
with RNC. Following all processing steps a mono-exponential model was fitted to the envelope of the data. With RNC the fitted parameters are $E_0 = 102 \pm 6$ nV and $T_2^* = 193 \pm 11$ ms which are close to the parameters of the synthetic NMR signal. If the RNC step is excluded, the envelope is more noisy and the fitted parameters are $E_0 = 132 \pm 16$ nV and $T_2^* = 150 \pm 23$ ms. These results show that RNC can be performed with the new instrument and improves SNR and accuracy of the estimated parameters.

**4.4   Effective dead time**

The effective dead time consists of instrument dead time and filter settling time during which the NMR signal is transiently distorted. In standard surface-NMR instruments where the same coil is used for both transmit and receive, the instrument dead time is caused by the switch procedure between the two modes. Apsu continuously samples the Rx coil signals without any switching and the instrument dead time is therefore potentially zero. However, the analog and digital filters affect the



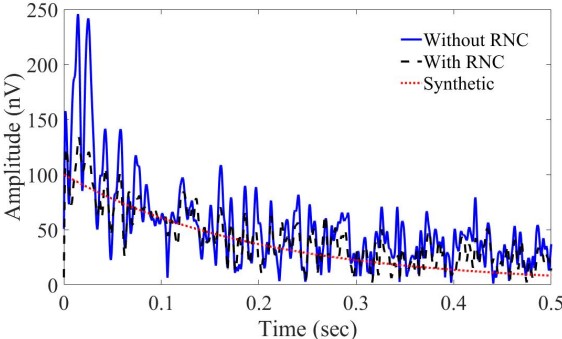

**Figure 8.** Envelopes extracted with and without RNC technique.

early times of the signal causing transient effects. Apsu uses untuned Rx coil and short settling-time bandpass filters between 1 kHz and 5 kHz which give a filter settling time of 0.42 ms. More filter settling time is added by a subsequent digital signal processing chain where SNR is increased with a digital bandpass filter.

We measured the effective dead time of the receiver system using the data collected at the Schillerslage test site in Hannover, Germany. The results are shown in Fig. 9. In the top panel the 40 ms long transmitter pulse with a 10 A peak current is shown along with the real and imaginary NMR envelopes (blue/black). The pulse starts at 51.2 ms and ends at 91.2 ms. After the transmitter shut off, the pulse current decays to 0 A in 1.8 ms. The NMR envelopes were obtained by stacking of 28 records and digital quadrature demodulation followed by low-pass filtering with a fourth-order digital Butterworth filter with

a cutoff frequency of 500 Hz and a settling-time of 3.6 ms. To preserve the signal initial phase during filtering, a forward and reverse filter was used. The NMR envelope is easily evident but an initial transient is also seen. The zoom-in view of the transmitter pulse turn-off in Fig. 9(b) shows that the transmitter current has dissipated to 0 A at 93.0 ms and that the NMR signal envelopes are distorted by the transients until approximately 97 ms. This gives an effective dead time of 5.8 ms in agreement with excitation current decaying time and the settling time of the analog and digital filters.

## 15   5   Field measurement

To validate the reliability of the receiver system for groundwater investigations, a field measurement was conducted at the Schillerslage test site near Hannover, Germany. The geological properties of the site have been characterized in previous studies (Müller-Petke et al., 2011; Grombacher et al., 2016). Lithological logging shows that the test site is characterized by two sandy aquifer layers. The water table is typically 1.5 m - 3 m below the surface and partially fills the upper aquifer consisting of

medium to coarse sands with inter-bedded peat layers down to about 11 m. A glacial till with fine sands separating the upper aquifer from the lower is about 6 m thick. The lower aquifer consisting of medium sands is about 3 m thick and bedrock is found beneath these layers.



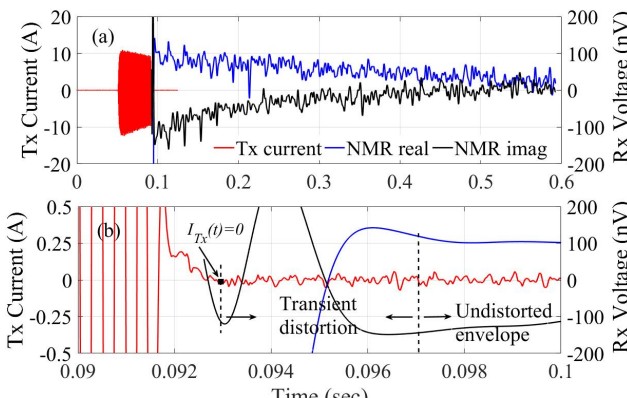

**Figure 9.** Excitation pulse (red) and the real (blue) and imaginary (black) parts of the surface-NMR signal envelopes. (a) Overview of excitation pulse and surface-NMR signal. (b) Zoom at the early-time signal.

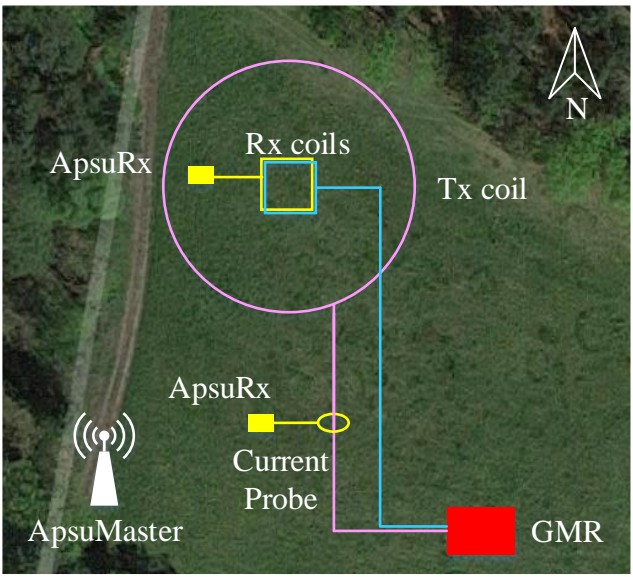

**Figure 10.** Field measurement setup with Apsu and GMR at the Schillerslage test site near Hannover, Germany.

The field setup is illustrated in Fig. 10. A commercial surface-NMR instrument, the GMR, Vista Clara, was used for comparison. The GMR had a dual purpose as it is a well-established instrument to compare with and it also supplied the transmit pulses needed for our receiver system. The GMR excitation pulses were transmitted using a 60 m diameter circular Tx coil (light purple circle) and the NMR signals were recorded with a 10 m by 10 m, 12-turn square Rx coil (light blue square, area-turn product 1200 $m^2$) located at the center of the Tx coil. In the Apsu receiver system, one channel in a ApsuRx was connected to a 10 m by 10 m, 8-turn square Rx coil (yellow square, area-turn product 800 $m^2$) overlapped with the GMR





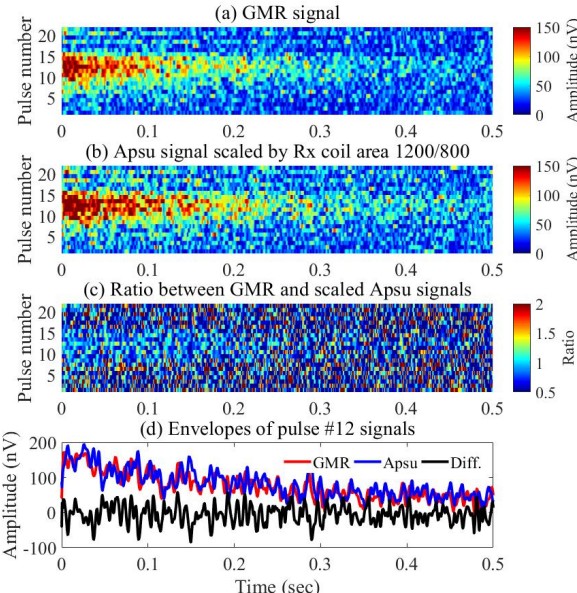

**Figure 11.** Data space comparison between the commercial GMR instrument and our Apsu receiver. (a) Data space of GMR results. (b) Data space of Apsu results after being scaled by the area-turn factor of 1200/800. (c) Ratio between the GMR and the Apsu signals. (d) Envelopes of the pulse #12 NMR signal from the two instruments and the difference.

Rx coil. One channel of another ApsuRx unit was connected to a high bandwidth current probe (yellow ellipse) to detect the turn-on time of the GMR transmitter. The ApsuMaster shown in the lower left controlled the data acquisition and retrieval in the ApsuRx units through wireless communication. The Larmor frequency at the site was 2104 Hz. In the measurements, 22
pulse moments sampling the interval from 0.09 As to 10 As was used all with a 40 ms pulse duration. Each pulse moment was repeated 28 times for stacking.

The signals recorded by two receivers were processed with the same procedures: band-pass filtering with a 500 Hz bandwidth, powerline harmonics subtraction with the model-based method, quadrature detection and stacking (Müller-Petke et al., 2016). Apsu data are further scaled by the 1200/800 ratio of the Rx coils area-turn product. The comparison of the extracted data
space from the two instruments is shown in Fig. 11. The initial signal amplitudes recorded by Apsu and GMR are higher than 100 nV within pulse number 10-15 (yellow zone) and are smaller than 50 nV in other pulse numbers (blue zone). Fig. 11(c) shows the ratio between the Apsu and GMR signal amplitudes for each point in the data space. The ratios are close to one in the region with strong signal level showing that the two instruments measure the same signal. Outside the region, the ratio is highly varying as these points in the data space contain no signal but only random noise, uncorrelated between the two receiver
systems. Fig. 11(d) shows the envelopes of the Apsu (blue) and GMR data (red) and their difference (black) from pulse number 12. The GMR and Apsu envelopes are effectively coincident, decaying from 180 nV with the same relaxation rate. The



difference between the two envelopes is random noise and no signal residual is visible. The same behavior is found in the other pulse moments.

An analysis of the noise in the data sets from the two instruments showed that externally coupled noise is the main component

in both cases. The Rx coils for GMR and Apsu were overlapped, making the noise levels in the two data sets almost similar and the small differences can be ascribed to the dissimilarities in receiver bandwidth of the two instruments.

## 6   Conclusions

We presented a new multichannel surface-NMR receiver system where ApsuRx units connecting up to three receiver coils are wirelessly connected to the ApsuMaster. The aim of the receiver system is to improve SNR and reduce measurement times for

multi-coil survey designs in groundwater investigations. We demonstrated that the wireless network performed as designed and remote reference noise cancellation can easily be carried out. In this proof-of-concept, a distance of 200 m between two receiver coils was used, but the distance can be increased several-fold and multiple reference coils can be employed. We showed that surface-NMR measurements can be performed with the new receiver system and in particular that the amplitude and relaxation rate of measured free induction decays are in complete agreement with measurements obtained with a commercial surface-

NMR instrument.

The wireless connections between the ApsuMaster and the ApsuRx allows us to place the reference receiver coils in proximity of identified noise sources. The actual gain in SNR obtained with this new field strategy is heavily dependent on the site-specific noise conditions. The optimization of SNR with the wireless receiver system is a research question which is currently being addressed.

*Competing interests.* The authors declare that they have no conflict of interest.

*Acknowledgements.* Simon Ejlertsen, and Bo Bjerre are gratefully acknowledged for their contributions in designing and building the instrument. Gordon Osterman is acknowledged for his review of an early version of the manuscript. We would also like to thank Mike Müller-Petke and Raphael Dlugosch for the GMR field data acquisition. The COWI foundation and Aarhus University Research Foundation (AUFF-E-2015-FLS-9-13) are acknowledged for providing financial support. Lichao Liu is supported by a grant from the China Scholarship Council. Denys Grombacher is funded by the Danish Council for Independent Research (DFF-5051-00002).





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
