# Peer review of "Apsu: a wireless multichannel receiver system for surface-NMR groundwater investigations"

_Geoscientific Instrumentation, Methods and Data Systems, 2018_

## Referee Comment (RC1) · R. Dlugosch (Referee) · 31 May 2018

General comments

1. Does the paper address relevant scientific questions within the scope of GI?

Yes, the paper present a new wireless multichannel receiver system for surface-NMR which was developed to increase the flexibility and efficiency for 3d SNMR surveys and remote reference noise cancellation.

2. Does the paper present novel concepts, ideas, tools, or data?

Yes the paper present a novel tool to record SNMR data. The novel instrumental concepts are a) differential Rx coils, b) dual acquisition of two gains, and c) wireless connection of Rx units. Especially the wireless connection of Rx units will improve the effective application of established survey and noise cancellation concepts and will trigger novel concepts

3. Are substantial conclusions reached?

The substantial conclusions should be modified. The "improve" in SNR, which is a proposed goal of the system development (P1/L3; P16/L9), is not explicitly shown.

- The noise properties of the system are provided in detail but are not compared to available systems therefore it is not possible to judge if there is any improvement. Also the authors seem to compare several features of Apsu with a NUMIS system (tuned Rx coils, non-continuous RX record) while there are other Rx+Tx systems available which already have this features for more than 10 years (GMR (Vista Clara, Walsh 2008), MIDI (Radic Research, Radic 2006)). I suggest to change the statement "to develop an SNMR instrument with high SNR"

- The noise cancellation using reference loop is shown (synthetic) but is not new in SNMR (GMR (Vista Clara, Walsh 2008), MIDI (Radic Research, Radic 2006)).

- The benefit of Apsu to be able to place a remote reference loop close to a noise source is not shown or referenced (it is mentioned in the outlook: P14/L16ff). Additionally, I have some doubts if this will work as intended. From my understanding of RNC, putting the Ref far away (several 100 m or up to 1km (P9/L4)) from the Rx, will generally reduce the correlation of the noise measured in both loops, which is essential for RNC. Therefore I would generally suggest to place the Ref as close to the Rx coil as possible without recording (= cancelling) NMR signal, not far away. Please find a reference supporting your findings or do not exaggerating the benefit of far separated Rx and Ref loops for RNC without any proof that this increase SNR.

- The wireless connected ApsuRx makes using multiple Ref loops very simple. However, the benefit of using multiple Ref loops is neither presented nor referenced (e.g.

Daalgard et al 2012?; Müller-Petke & Costabel 2014). One simple way to show the benefit with the presented data might be to provide subsequent RMS values after RNC(Ref1) and RNC(Ref2)

- The new feature of dual recording using two gain factors to reduce the chance of data clipping is well presented in the paper but the benefits are neither shown nor referenced.

- The concept and benefit of using differential Ref. coils for SNMR application is also neither shown nor referenced. Please at least provide a reference for its success in another EM method (TEM?)

4. Are the scientific methods and assumptions valid and clearly outlined?

Yes

5. Are the results sufficient to support the interpretations and conclusions?

Generally yes.

6. Is the description of experiments and calculations sufficiently complete and precise to allow their reproduction by fellow scientists (traceability of results)?

Yes the descriptions are generally very detailed and the data is very well presented

7. Do the authors give proper credit to related work and clearly indicate their own new/original contribution?

P2/L24-26: I miss the reference to an existing and commercially available instrument (MIDI, Radic Research (e.g. Radic 2007)) that already features a separated Tx and Rx loop wire and therefore already do continuous Rx records with untuned coils

P4/L7: I miss a reference that several new SNMR instruments already use untuned Rx coils (GMR Walsh 2008, MIDI Radic 2006) for more than 10 years, therefore it is not a new Apsu feature. Additionally, the discussion about the properties of tuned Rx coils

seem to be motivated by the comparison with a NUMIS system. Since this does not present the state-of-the-art it could easily be shortened.

8. Does the title clearly reflect the contents of the paper?

Yes

9. Does the abstract provide a concise and complete summary?

Yes

10. Is the overall presentation well structured and clear?

Yes

11. Is the language fluent and precise?

I am not a native speaker myself but the text appears mostly well written. However, there are sentences which seem to lack a proper conjunction e.g. repeatedly starting with "The..." several times.

12. Are mathematical formulae, symbols, abbreviations, and units correctly defined and used?

Generally yes. Some minor remarks are provided in the "technical corrections"

13. Should any parts of the paper (text, formulae, figures, tables) be clarified, reduced, combined, or eliminated?

See "technical corrections". Some parts e.g. referring to tuned coils, might be reduced

14. Are the number and quality of references appropriate?

Generally yes

15. Is the amount and quality of supplementary material appropriate?

Yes

[Figure]

Scientific questions/issues

General: Details on the electronics are beyond my expertise!

P2/L14: as I have mentioned in the general section, I am missing the proof or a reference that shows that a Ref loop should be placed far away from the Rx and close to the noise source. Here would be a good place to provide such a reverence.

P4/L19: What are "differential coils" and what are their benefit? Reference them (at least their success in other EM methods) or describe them and show that they can consistently improve SNR for SNMR applications. It is an interesting but after my knowledge unproven concept for SNMR.

P11/16ff: Reference noise cancellation

The RNC scheme is not new (Radic 2006, Walsh 2008). The benefit of Apsu is that the Ref loop can be placed without a wire connection, i.e. quickly and far away from the Tx/Rx. The arising questions are:

a) is the timing i.e. synchronisation jitter between the units, small enough to use RNC? -> seems perfectly fine

b) Is there a benefit of using multiple Ref loops (since they are easy to lay out) recording different noise characteristics. Not presented, but could easily be shown by subsequent RNC using both ref datasets and providing the respective improvement in RMS

c) Is there no harm for the RNC to increase the distance between Rx and Ref to 200m (or up to 1km)? Sadly this is not shown and would require additional experiments. Maybe the authors find a reference to proof this or they should significantly soften their statement that this improves SNR and discuss the drawbacks.

P12/L13-15: I am a little confused by the provided SNR of the envelops in dB (P12/L14)? How is SNR in dB calculated? SNR = 10*log10(100nV/RMS) for amplitudes? Therefore 0.4 dB = 100/91 [Sig/RMS] whereas 5.1 dB = 100/31 [Sig/RMS]?

RMS with RNC RMS w/o RNC

Filtered 147 147

HNC 85 85

RNC 62 -

18 times stacking (sqrt(18) $\sim$ 4) 31 91

Something is clearly wrong here! (maybe I am) The noise increases w/o RNC after stacking? Please provide the SNR not only in dB but additionally the RMS value of the noise (or the data misfit) after stacking. Also the achieved reduction of the noise due to stacking is expected to be close to 4, not 2 (w RNC) or even <1 (w/o RNC).

Technical corrections

P1/L12: please make the numbers consistent. The effective dead time of the ApsuRx (including filtering) should be 3.6 ms (+0.4ms?). In the presented example, the distorted section of the NMR record (including Tx effects) is 5.8 ms (which you confusingly also call effective dead time).

P1/L20: check the author guidelines if you need to introduce the acronyms (Surface-NMR) again after the abstract. The same is true for SNR (P2/L9)

P1/L21: I do not think that Lehman-Horn et al 2012 is an appropriate reference for SNMR and aquifer properties. Please check if you find a better suited reference

P2/L14: What is a primary coil? Maybe introduce this phrase at P2/L11 as Tx+Rx which you later adapt to primary channel

Figure 1: ". . . multichannel surface-NMR receiver system Apsu." Since the system does not allow for Tx (yet). I suggest avoiding any misunderstandings and being consistent to the paper title.

P3/L3: "GPS time signal" instead of "clock"?

P3/L5: provide country for Vista Clara Inc.

P3/L5: What is a "primary channel"? Similar to the previous "primary coil" the definition is unclear

P3/L12: how are the channel "configured"? Please provide some additional information like e.g. "recording parameter etc."

P3/L13: "is connected to an array"

P4/L3-7: the described amplification of Rx using a tuned coil is not state-of-the-art. The description could be shortened and the currently favoured concept of using untuned Rx coils should be presented (Walsh 2008, Radic 2006)

P4/L22: "... into the critically or slightly over-damped state."

P4/L30: check the consistent use of excitation or transmit pulse in the paper

P4/L32: I was a little confused by this sentence. Maybe us " ... resistor...prevents any induced current in the Rx loop due to the Tx pulse which can disturbed the magnetic excitation field"

P4/L34: twisted and shielded cable

P5/L8 (Eq. 2) provide $\omega$ or introduce it earlier in the text

Figure 4: The acronym AP for access point is explained way after the first reference to Fig. 4. Just write it out

P8/L19: "The GPS module only needs the signal from a single satellite..."

P8/L24-25: the acronyms for synchronization time and time stamp were both already introduced in P8/L22. Just use either the acronyms or the words here.

P8/L30: Provide \lambda in Eq.7? time shift per passed time?

P9/L10, L11 and L17 Consider using "(Fig. X)" instead of "..., Fig. X".

P9/L13-14: What do you mean with this sentence? I am quite sure that I misunderstood this. Is each ApsuRx connected to a single and specific WiFi antenna respectively? If that is the case, you could only connect up to 8 ApsuRx? And you would need to place the Apsu Master in the centre of the layout since every antenna has a limited angle of view?

P9/L15-16: Please check the authors guideline but I think you generally skip the blank between X and degree = X°

P9/L17: "provide power" might be misleading " forward"?

P10/L1: " to the AP in the WiFi tower..."

P10/L29: Spell out "Section 2.1"

Figure 6: "Scatter plot of one second of recording from two channels with shortened ..." The following sentence about bin width and number does not provide any significant information and could be deleted. The red line is very thin and barely visible.

P11/L8 (Eq 9) provide $\omega$ or introduce it earlier in the text

P11/L17: "We tested the applicability of a reference noise cancellation (RNC) scheme with wireless ..."

Figure 7: The red squares are very small and barely visible. Please increase the size of the data points and maybe reduce the number of data points if they are redundant (or provide their STD instead)

Figure 8: The most important line has low variations and is dashed and therefore can hardly be seen. Please consider to flip the line style and show the w/o RNC as a dashed line The caption is eye-catchingly short compared to other figure captions and lack information. E.g. add that these are envelopes of an NMR signal to show the performance of RNC etc.

P12/L3: The loop layout of the RNC experiment is not clearly described. Only the

distance of the ApsuRx to Rx is provided (200m) and the distance between both Ref (100m). The lacking information is the distance Rx to Ref for both Ref loops. A small sketch might help if the layout is too complex to describe.

P12/L8ff: Can you please provide a reference for this typical SNMR processing scheme

P13/L3 ". . . which leads to a filter. . ."

P13/L4 the arising question is how much filter settling time is added (which is answered a few sentences later). But maybe add a comment like or "e.g. 3.6 ms for a 500 Hz butterworth filter" and lead over to the next passage by "an example is provided in the following"

P13/L5 ". . . using data collected. . .test site near Hannover" The field example is not yet presented in the paper

P13/L7ff (also Fig 9) please consider to shift the (arbitrary chosen) time axis to t=0 at the end of the pulse which makes the (overall very nice) figure and times easier to read. Many times you provide to need to be subtracted by 91.2ms to be of relevance.

P13/L9: ". . . quadrature detection. . ." both is true but stick to one term during the paper

P13/L14: See also abstract. You are not consistent when you talk about the effective deadtime. In the abstract you refer to 4ms (3.6ms + 0.42ms? P13/L3+10) which is only the Rx filtering and here you include the artefact due to excitation current decay (5.8ms). Personally, I think that ApsuRx has an effective dead time of ∼4ms but dependent on the used Tx you should clip the data to 6ms to avoid pulse artifacts. Once Apsu includes a Tx you should provide the maybe longer effective deadtime for the whole SNMR system. Please consider to avoid calling it "effective deadtime" here and change the sentence to "5.8 ms including excitation current decay. . ."

P14/L3: ". . .well-established surface-NMR Rx system. . . "

P14/L6: The Apsu receiver system might be misleading as you presented ApsuRx.

Maybe introduce the System consisting of one Apsu Master and two ApsuRx first. E.g. "The used Apsu receiver system consists of one Apsu Master and two ApsuRx, One channel of an ApsuRx was . . . ."

Figure 11: "after being scaled to the GMR signal by the area-turn factor of the coils (1200/800)"

P15/L7: "The signal recorded by the two Rx instruments (GMR, Apsu) were processed. . . "

P16/8: " . . .receiver system where multiple Apsu Rx units each connecting . . . connected to an ApsuMaster"

P16/9: see previous comments on improving SNR. You do not compare the SNR properties of your system to another system. While Apsu might be a significant upgrade to your NUMIS, the in detail presented features to improve SNR (RNC, short dead time) are state-of-the-art (GMR, MIDI Radic). The impact of the new features (Wireless connection (outlook), dual gain recording, differential coils+ Rx) to improve SNR are not shown. Please simply rephrase it to ". . . the aim of the receiver system is a high SNR and . . ."

P16/11+16ff: see previous comments on widely separated Rx and Ref loops. Please add a comment that modify this statement. While a long distance between Rx and Ref loops is technically possible with Apsu, I have strong doubts that RNC will perform well or even improve

Please also note the supplement to this comment:
https://www.geosci-instrum-method-data-syst-discuss.net/gi-2018-1/gi-2018-1-RC1-supplement.pdf

---

## Referee Comment (RC2) · PhD Irons (Referee) · 13 Jun 2018

The authors present a fairly comprehensive description of a new modular surface NMR receiving instrument. The wireless nature of the system is novel and should open up survey design. Additionally, a more open instrument description is welcome compared to commercial 'black-box' systems. As such, this paper presents a relevant and important contribution to the surface NMR literature.

1. *Does the paper address relevant scientific questions within the scope of GI?* Yes, the topic is very relevant to *Geoscientific Instrumentation*.

2. *Does the paper present novel concepts, ideas, tools, or data?* Yes, this paper

describes the first wireless distributed receiver sensor network for surface NMR surveys.

3. *Are substantial conclusions reached?* Yes, the authors have successfully designed, built, and deployed a seemingly field-rugged system utilizing GPS timing synchronisation. This is no small feat. Bravo!

4. *Are the scientific methods and assumptions valid and clearly outlined?* Yes, the system is described in a fair degree of detail. A better description or citation of differential coils should be included. I would also like to see a description of the power requirements of the receivers, and how long data collection can be performed on a single charge.

5. *Are the results sufficient to support the interpretations and conclusions?* The authors claim to have improved upon the SNR of the measurements. However, the field example does not demonstrate a reduced noise floor compared to other available instrumentation. If the authors should substantiate this statement, or remove it from the manuscript.

6. *Is the description of experiments and calculations sufficiently complete and precise to allow their reproduction by fellow scientists (traceability of results)*? This is a tricky question. This manuscript would not be sufficient to replicate their design. However, the authors do not elude to this being an open-source design. That said, the description is sufficient for *users* of the instrument to gain a much better understanding of system noise, response, and related issues. This information is invaluable for calibration, as such, I find this aspect of the paper acceptable and beyond what many instrument manufacturers provide.

7. *Do the authors give proper credit to related work and clearly indicate their own new/original contribution?* On page 2, line 10 a description of the first surface

NMR instruments cites Legchenko and Valla, 2002 which describes the Iris NU-MIS. However, the first surface NMR instrument was the Hydroscope:

```
@INPROCEEDINGS{Semenov1987,
  author = {A. G. Semenov},
  title = {{NMR} hydroscope for water prospecting},
  booktitle = {Expanded Abstracts},
  year = {1987},
  pages = {66--67},
  organization = {Indian geophysical Union},
  note = {Proceedings of the Seminar on Geotomography, Hyderabad}
}
```

In this circumstance, it would be appropriate to cite the first instrument in addition to the NUMIS.

8. *Does the title clearly reflect the contents of the paper?* It does, however the acronym(?) 'Apsu' is never defined. If it has some sort of meaning, please define it in the copy somewhere.

9. *Does the abstract provide a concise and complete summary?* It does, however the discussion of SNR improvements either need to be substantiated or removed from the abstract as well.

10. *Is the overall presentation well structured and clear?* The paper is well structured, with the exception of §4.3 §4.4 and §5. The dead time discussion and (to some extent) field noise synthetics follows from the field examples. It would be more clear to introduce the field cites as e.g. 'Field Validations' with subsections dedicated to dead time realizations and noise synthetics. As it stands Schillerslage is introduced twice and Silkeborg once. If the authors want to keep the Silkeborg examples in §4 that would be fine, but I would still recommend moving the

dead time to §5 with a new §5.2 describing the data comparisons. If the phase is presented (discussed below), this could be a separate section as well.

11. *Is the language fluent and precise?* The manuscript is well written. On page 10 line 5 a trailing apostrophe (') is used where a leading apostrophe (') should be.

12. *Are mathematical formulae, symbols, abbreviations, and units correctly defined and used?* Yes.

13. *Should any parts of the paper (text, formulae, figures, tables) be clarified, reduced, combined, or eliminated?* See above discussion of §4 and §5.

14. *Are the number and quality of references appropriate?* While the use of GPS timing is novel in surface NMR, it is common in the MT/CSEM community. For example the Zonge Zen system. A citation of this prior art would be appropriate and also affirm that GPS timing can reliably be used. Seismic nodal systems also use GPS timing and can be cited.

15. *Is the amount and quality of supplementary material appropriate?* N/A

In addition to the points above, I offer a few additional suggestions for consideration.

- The use of the word 'identical' on P. 3 line 5 to describe the recorded NMR signals should be avoided. This description gives the impression that the two signals have no discernable measure between. 'Practically equivalent' or some similar verbiage would be preferable.

- The jet colourmap in figure 11 should be replaced with a perceptually uniform one. Additionally, the colormap clips at 0, but the quadrature detection should result in negative values as well. A diverging colormap centred around 0 is highly encouraged for the top two subfigures.

- Complex inversion is an important consideration in surface NMR, especially with separated transmitters and receivers. Data phase comparisons (or real/imaginary plots) with the GMR are highly encouraged and will confirm that the developed instrumentation is at the 'bleeding edge' of surface NMR instrumentation.

---

## Author Comment (AC1) · 4 Jul 2018

**Response to Referees' Comment on "Apsu: a wireless multichannel receiver system for surface-NMR groundwater investigations" by Lichao Liu et al.**

Lichao Liu et al.
lichao@geo.au.dk

To referee R. Dlugosch (gi-2018-1-RC1):

Dear Referee, Thank you for reviewing our manuscript and raising issues that help to improve the manuscript. The authors response to all the comments in the following context and a marked-up manuscript is appended.

**1 Response to General comments**

1) **Comment 3:** The substantial conclusions should be modified. The "improve "in SNR, which is a proposed goal of the system development (P1/L3; P16/L9), is not explicitly shown.

- The noise properties of the system are provided in detail but are not compared to available systems therefore it is not possible to judge if there is any improvement. Also the authors seem to compare several features of Apsu with a NUMIS system (tuned Rx coils, non-continuous RX record) while there are other Rx+Tx systems available which already have this features for more than 10 years (GMR (Vista Clara, Walsh 2008), MIDI (Radic Research, Radic 2006)). I suggest to change the statement "to develop an SNMR instrument with high SNR ".

  **Response:** Agree, the main features of our wireless receiver are increased Rx loops deployability and reduced effort in field measurements. The collected data is normally dominated by the couping EM noise. The actual gain in SNR obtained with this new field strategy is heavily dependent on the site-specific noise conditions. But it is potential to improve SNR in some scenarios which will be demonstrated in detail later. Due to the practical reason, we cannot compare the developed receivers with the existing system directly.

  **Change in the manuscript:** The statement *to develop an SNMR instrument with high SNR* is removed in the abstract.

- The noise cancellation using reference loop is shown (synthetic) but is not new in SNMR (GMR (Vista Clara, Walsh 2008), MIDI (Radic Research, Radic 2006)).

  **Response:** Indeed. The result is shown to demonstrate that the developed receivers are synchronous with the wireless connection and is capable of reference noise cancellation.

- The benefit of Apsu to be able to place a remote reference loop close to a noise source is not shown or referenced (it is mentioned in the outlook: P14/L16ff). Additionally, I have some doubts if this will work as intended. From my understanding of RNC, putting the Ref far away (several 100 m or up to 1km (P9/L4)) from the Rx, will generally reduce the correlation of noise measured in both loops, which is essential for RNC. Therefore I would generally suggest to place the Ref as close to the Rx coil as possible without recording (= cancelling) NMR signal, not far away. Please find a reference supporting your findings or do not exaggerating the benefit of far separated Rx and Ref loops for RNC without any proof that this increase SNR.

  **Response:** It is true that the correlation between two Rx coils increase when the their distance decreases. It is better to place the Ref coil as close to the signal coil as possible if the data is dominated by far field noise sources, e.g. the noise originated form atmospheric. But in the case when multiple noise sources are

[Figure]

Figure 1: An example of normalized amplitude of data recorded by the dual-gain channel of Apsu.

in the proximate to signal loop. The pathways from multiple noise sources to a Rx coil are not identical. The transfer functions to estimate the noise components in the signal coil from different noise sources are divergent Larsen et al. [2014]. If there are multiple noise present in a Ref coil, the filters computed by adaptive or Winer algorithms will be the overall result rather than transfer functions optimum for each sources. Hence, the RNC efficiency can be improved if each reference coil just or mainly record one specific noise sources.

– The wireless connected ApsuRx makes using multiple Ref loops very simple. However, the benefit of using multiple Ref loops is neither presented nor referenced (e.g. Daalgard et al 2012?; Müller-Petke & Costabel 2014). One simple way to show the benefit with the presented data might be to provide subsequent RMS values after RNC(Ref1) and RNC(Ref2).

**Response:** The proposal of providing the subsequent noise levels after the Ref1 and Ref2 are adopted. The RMS values are 67 nV after the RNC with Ref1 and 62 nV after the RNC with both Ref1 and Ref2. Also, multiple Ref loops is beneficial in scenario when multiple noise sources are in presence as shown above,.

**Change in the manuscript:** The sentence *Reference noise cancellation (RNC) using Wiener filtering and noise from two reference coils further reduced the RMS value to 62 nV* is revised to *Reference noise cancellation (RNC) reduced the RMS value to 67 nV using noise from one reference coil and to 62 nV with noise from two reference coils.*

– The new feature of dual recording using two gain factors to reduce the chance of data clipping is well presented in the paper but the benefits are neither shown nor referenced.

**Response:** High-gain is beneficial to record to nano-voltage level NMR signal, especially the small coil is employed in our system. At the beginning of one measurement, the gain factor is configured but the noise level could increase later. In this case, the high-gain channel may be saturated but the low-gain channel is still usable, shown in Figure 1. The recorded signals with amplitude lower than the limit of high-gain channel are identical but inly the data recorded by the low-gain channel is usable when the signal amplitude exceed the limit of high-gain channel. There are already four performance verification results are presented, hence this figure is not shown.

– The concept and benefit of using differential Ref. coils for SNMR application is also neither shown nor referenced. Please at least provide a reference for its success in another EM method (TEM?)

**Change in the manuscript:** Two references (Nyboe and Sørensen [2012], Chen et al. [2015]) demonstrate the benefits of the differential coil in the airborne EM system are added. And the sentence *Differential Rx coil is beneficial to cancel common-mode noise Nyboe and Sørensen [2012], Chen et al. [2015]* is added.

2) **Comment 7:**

– P2/L24-26: I miss the reference to an existing and commercially available instrument (MIDI, Radic Research (e.g. Radic 2007)) that already features a separated Tx and Rx loop wire and therefore already do continuous Rx records with untuned coils.

**Change in the manuscript:** The reference (Radic [2006]) related to MRS-MIDI, Radic Research, Germany is now cited in the Rx coil section.

– P4/L7: I miss a reference that several new SNMR instruments already use untuned Rx coils (GMR Walsh 2008, MIDI Radic 2006) for more than 10 years, therefore it is not a new Apsu feature. Additionally, the discussion about the properties of tuned Rx coils seem to be motivated by the comparison with a NUMIS system. Since this does not present the state-of-the-art it could easily be shortened.

**Change in the manuscript:** The references (Radic [2006], Walsh [2008]) related to untuned Rx coils have been recited there. However, the construction of these coils are quite different from the construction used in Apsu.

**Change in the manuscript:** A sentence is added: *The GMR and MRS-MIDI employ the untunned coil Radic [2006], Walsh [2008].*

**2   Response to Scientific questions/issues**

1) **Comment:** P2/L14: as I have mentioned in the general section, I am missing the proof or a reference that shows that a Ref loop should be placed far away from the Rx and close to the noise source. Here would be a good place to provide such a reverence.

   **Response:** Please refer to the response in the General comment 3.

2) **Comment:** P4/L19: What are "differential coils "and what are their benefit? Reference them (at least their success in other EM methods) or describe them and show that they can consistently improve SNR for SNMR applications. It is an interesting but after my knowledge unproven concept for SNMR.

   **Response:** Two references Nyboe and Sørensen [2012], Chen et al. [2015] addressed the differential Rx coil in airborne TEM are cited. Compared with the traditional coil, the differential coil has three output end: the positive, ground and negative.

   **Change in manuscript:** The sentence *Differential Rx coil is beneficial to cancel the common-mode noise and it is able to reduce the Johnson noise of the coil by half [Nyboe and Sørensen, 2012, ?]. The typical common mode noise is the induced noise in the leading cable and the wiring of the acquisition board by the coupling noise.* is added.

3) **Comment:** P11/16ff: Reference noise cancellation The RNC scheme is not new (Radic 2006, Walsh 2008). The benefit of Apsu is that the Ref loop can be placed without a wire connection, i.e. quickly and far away from the Tx/Rx. The arising questions are:

   a) is the timing i.e. synchronization jitter between the units, small enough to use RNC? → seems perfectly fine

      **Response:** The jitter is determined by the GPS and is approximately 20 ns which is way more accurate than what is needed. The recordings from two receiver box with a distance of 250 m away are shown in Figure 2. We can find four spikes in signal coil and Ref coil happened at the same time stamp and have the same duration which confirm the two wireless channel is synchronous with ignorable jitter.

   b) Is there a benefit of using multiple Ref loops (since they are easy to lay out) recording different noise characteristics. Not presented, but could easily be shown by subsequent RNC using both ref datasets and providing the respective improvement in RMS.

      **Response:** It is beneficial to layout multiple Ref coils when multiple noise sources are presented. The RMS vales 85 nV before RNC, and decrease to 67 nV and 62 nV with one and two Ref coils.

   c) Is there no harm for the RNC to increase the distance between Rx and Ref to 200m (or up to 1km)? Sadly this is not shown and would require additional experiments. Maybe the authors find a reference to proof this or they should significantly soften their statement that this improves SNR and discuss the drawbacks.

      **Response:** As described below, the Ref coil will be placed as close to the signal coil as possible when there is no knowledge of the noise distribution. In case the locations of noise sources are know, for instance power lines, electric fences, human installations, it is better to employ the Ref coils close to the noise sources, even it is maybe 200 or 300 m away from the signal loop. The RNC efficiency can be improved when each Ref coil records a specific noise. The reference coils is within hundreds meters away from the Tx loop and hence the aimed wireless coverage is 1 km.

[Figure]

Figure 2: Recordings by two receiver boxes with a distance of 250 m. Four spikes are presented in both of the coils at the same time.

4) **Comment:** P12/L13-15: I am a little confused by the provided SNR of the envelops in dB (P12/L14)? How is SNR in dB calculated? SNR = 10*log10(100nV/RMS) for amplitudes? Therefore 0.4 dB = 100/91 [Sig/RMS] whereas 5.1 dB = 100/31 [Sig/RMS]? RMS with RNC RMS w/o RNC

| Filtered | 147 | 147 |
| HNC | 85 | 85 |
| RNC | 62 | - |

18 times stacking (sqrt(18) 4) 31 91 Something is clearly wrong here! (maybe I am) The noise increases w/o RNC after stacking? Please provide the SNR not only in dB but additionally the RMS value of the noise (or the data misfit) after stacking. Also the achieved reduction of the noise due to stacking is expected to be close to 4, not 2 (w RNC) or even <1 (w/o RNC).

**Response:** Due to the amplitude of NMR signal decays but the noise is stationary, which means we cannot only use the initial amplitude 100 nV as the RMS value of the signal. Hence, the SNR of an envelope in the context is calculated as,

$$SNR = 10\log\Big(\frac{\sum_{k=0}^{N}(E_0 e^{-t(k)/T_2^*})^2}{\sum_{k=0}^{N}(E(k) - E_0 e^{-t(k)/T_2^*})^2}\Big), \tag{1}$$

where $E$ is the obtained envelope with noise, $s_0 e^{-t(k)/T_2^*}$ is the synthetic signal and $N$ is the datum of envelope. The SNR envelope can describe the data quality of the retrieved envelope. The RMS values of noise with and without RNC after stacking are 18 nV and 30 nV, respectively. The ratio between RMS values after and before stacking without RNC is 30/62 is around 2 not 4 which is expected. That is because the noise in different stacks are not only un-correlated random noise. Only the RMS of random noise will be reduced to $\frac{1}{\sqrt{N_{stack}}}$.

**Change in the manuscript:** The SNR definition is added in the manuscript. The RMS value of the noise after stacking is added *The RMS values of noise with and without RNC after stacking are 30 nV and 18 nV, respectively.*

**3    Response to Technical corrections**

1) **Comment:** P1/L12: please make the numbers consistent. The effective dead time of the ApsuRx (including filtering) should be 3.6 ms (+0.4ms?). In the presented example, the distorted section of the NMR record (including Tx effects) is 5.8 ms (which you confusingly also call effective dead time).

   **Change in manuscript:** The effective dead time is changed to 5.8 ms in P1/L12 to make them consistent.

2) **Comment:** P1/L20: check the author guidelines if you need to introduce the acronyms (SurfaceNMR) again after the abstract. The same is true for SNR (P2/L9).

**Response and change in manuscript:** Surface-NMR and SNR have been defined in the abstract and then again at the first instance in the rest of the text.

3) **Comment:**P1/L21: I do not think that Lehman-Horn et al 2012 is an appropriate reference for SNMR and aquifer properties. Please check if you find a better suited reference.

   **Change in manuscript:** That reference is replaced by Legchenko et al. [2002].

4) **Comment:**P2/L14: What is a primary coil? Maybe introduce this phrase at P2/L11 as Tx+Rx which you later adapt to primary channel.

   **Change in manuscript:** The *primary coil* is replaced by *signal coil* in the context.

5) **Comment:**Figure 1: ". . . multichannel surface-NMR receiver system Apsu."Since the system does not allow for Tx (yet). I suggest avoiding any misunderstandings and being consistent to the paper title.

   **Change in manuscript:** *multichannel surface-NMR receiver system Apsu* is changed to *multichannel surface-NMR receiver Apsu.*

6) **Comment:**P3/L3: "GPS time signal"instead of "clock"?

   **Response:** GPS time is only output in the unit of second and it is the GPS clock maintain an accuracy on the order of nano-second.

7) **Comment:**P3/L5: provide country for Vista Clara Inc.

   **Change in manuscript:** The country is added.

8) **Comment:** P3/L5: What is a "primary channel "? Similar to the previous "primary coil"the definition is unclear

   **Response and change:** The *primary channel* is replaced by *signal channel* for better understanding.

9) **Comment:** P3/L12: how are the channel "configured "? Please provide some additional information like e.g. "recording parameter etc."

   **Change in manuscript:** Sentence *for instance the gain factors and synchronization time* is supplemented.

10) **Comment:** P3/L13: "is connected to an array "

   **Change in manuscript:** *is* is added.

11) **Comment:** P4/L3-7: the described amplification of Rx using a tuned coil is not state-of-the-art. The description could be shortened and the currently favored concept of using untuned Rx coils should be presented (Walsh 2008, Radic 2006)

   **Change in manuscript:** The references Radic [2006], Walsh [2008] have been cited.

12) **Comment:** P4/L22: ". . . into the critically or slightly over-damped state."

   **Change in manuscript:** *into the critically damped state or slightly over-damped* is changed to *into the critically or slightly over-damped state.*

13) **Comment:** P4/L30: check the consistent use of excitation or transmit pulse in the paper

   **Change in manuscript:** They are called as *excitation pulse* throughout the context.

14) **Comment:** P4/L32: I was a little confused by this sentence. Maybe us ". . . resistor...prevents any induced current in the Rx loop due to the Tx pulse which can disturbed the magnetic excitation field "

   **Change in manuscript:** The sentence *A current limiting resistor in series with the diodes eliminates the effect on the excitation magnetic filed by the induced current in the Rx loop.* is now removed.

15) **Comment:** P4/L34: twisted and shielded cable.

   **Response and change in manuscript:** *and shielded* is added.

16) **Comment:** P5/L8 (Eq. 2) provide $\omega$ or introduce it earlier in the text.

   **Response:** $\omega$ is described.

17) **Comment:** Figure 4: The acronym AP for access point is explained way after the first reference to Fig. 4. Just write it out

**Change in manuscript:** *access point* replaces the acronym AP.

18) **Comment:** P8/L19: The GPS module only needs the signal from a single satellite. . .

**Change in manuscript:** *The GPS module only needs single satellite to provide accurate pulse timing and it works nicely in practice, even with limited field of view and cloud cover* is changed to *The GPS module only needs the signal from a single satellite to provide accurate pulse timing and it works nicely in practice, even with limited field of view and cloud cover.*

19) **Comment:**P8/L24-25: the acronyms for synchronization time and time stamp were both already introduced in P8/L22. Just use either the acronyms or the words here.

**Response:** *synchronization time* and *time stamp* have been deleted.

20) **Comment:**P8/L30: Provide $\lambda$ in Eq.7? time shift per passed time?

**Response:** $\lambda$ is referred before in P8L14. It is the decimation ratio if the ADC.

21) **Comment:**P9/L10, L11 and L17 Consider using "(Fig. X) "instead of ". . ., Fig. X ".

**Response:** They are at the end of sentences, we think the period is necessary.

22) **Comment:**P9/L13-14: What do you mean with this sentence? I am quite sure that I misunderstood this. Is each ApsuRx connected to a single and specific WiFi antenna respectively? If that is the case, you could only connect up to 8 ApsuRx? And you would need to place the Apsu Master in the centre of the layout since every antenna has a limited angle of view?

**Response:** As shown in Fig. 5, each ApsuRx is connected to an antenna in the client mode. The ApsuMaster is connected to eight antennas in the access point mode and should be placed at the center of the layout. But each access point can be connected to multiple client antennas. Therefore, ApsuMaster can connect tens of or more ApsuRx.

23) **Comment:**P9/L15-16: Please check the authors guideline but I think you generally skip the blank between X and degree = Xdeg

**Response:** They are generated in LATEX, and should be in the right format.

24) **Comment:**P9/L17: "provide power"might be misleading "forward"?

**Response and changes in Manuscript:** You are right, the antenna can not provide power itself. *provides power* is replaced by *forwards power.*

25) **Comment:**P10/L1: "to the AP in the WiFi tower. . ."

**Changes in Manuscript:** *the* is inserted.

26) **Comment:**P10/L29: Spell out "Section 2.1"

**Response:** The abbreviation "Sect. "is required according to the journal's guidelines.

27) **Comment:**Figure 6: "Scatter plot of one second of recording from two channels with shortened . . ."The following sentence about bin width and number does not provide any significant information and could be deleted. The red line is very thin and barely visible.

**Changes in manuscript:** Because the bin width and number are related to the probability density values. Hence, the authors think it should be kept. The red line in Fig. 6 is bolded.

28) **Comment:**P11/L8 (Eq 9) provide $\omega$ or introduce it earlier in the text

**Changes in manuscript:** $\omega$ is described.

29) **Comment:**P11/L17: We tested the applicability of a reference noise cancellation (RNC) scheme with wireless...

**Changes in Manuscript:** Sentence is changed to *We tested the applicability of the reference noise cancellation (RNC) scheme ....*

30) **Comment:** Figure 7: The red squares are very small and barely visible. Please increase the size of the data points and maybe reduce the number of data points if they are redundant (or provide their STD instead)

   **Changes in Manuscript:** The red squares in Fig. 7 is larger.

31) **Comment:** Figure 8: The most important line has low variations and is dashed and therefore can hardly be seen. Please consider to flip the line style and show the w/o RNC as a dashed line The caption is eye-catchingly short compared to other figure captions and lack information. E.g. add that these are envelopes of an NMR signal to show the performance of RNC etc.

   **Response and changes in Manuscript:** The caption is added ... *to demonstrate the SNR improvements of RNC. The red dotted line is the synthetic envelope. The blue solid line and black dashed line are the results processed without and with RNC.*

32) **Comment:**P12/L3: The loop layout of the RNC experiment is not clearly described. Only the distance of the ApsuRx to Rx is provided (200m) and the distance between both Ref (100m). The lacking information is the distance Rx to Ref for both Ref loops. A small sketch might help if the layout is too complex to describe.

   **Change in manuscript:** *A second ApsuRx, located approximately 200 m away, served as the remote reference receiver and was connected to two Rx coils. All coils were 5 m by 5 m, 16-turn coils. The distance between the two reference coils was approximately 100 m* is replaced by  *A second ApsuRx, located approximately 200 m away in the east, served as the remote reference receiver and was connected to two Rx coils. All coils were 5 m by 5 m, 16-turn coils. Two Rx coils were located in the north and south of the reference receiver and the distance between the two reference coils was approximately 100 m.*

33) **Comment:**P12/L8ff: Can you please provide a reference for this typical SNMR processing scheme

   **Changes in Manuscript::** Müller-Petke et al. [2016] is added.

34) **Comment:**P13/L3 ". . . which leads to a filter. . ."

   **Changes in Manuscript:** *gives* is replaced by *leads to.*

35) **Comment:**P13/L4 the arising question is how much filter settling time is added (which is answered a few sentences later). But maybe add a comment like or "e.g. 3.6 ms for a 500 Hz butterworth filter and lead over to the next passage by an example is provided in the following "

   **Response:** *e.g. 3.6 ms for a 500 Hz butterworth filter* is added after the ... *with a digital filter.*

36) **Comment:**P13/L5 ". . . using data collected...test site near Hannover "The field example is not yet presented in the paper

   **Response and change in manuscript:** By adopting another referee's suggestion, the dead time subsection is moved behind the field measurements.

37) **Comment:**P13/L7ff (also Fig 9) please consider to shift the (arbitrary chosen) time axis to t=0 at the end of the pulse which makes the (overall very nice) figure and times easier to read. Many times you provide to need to be subtracted by 91.2ms to be of relevance.

   **Response:** The authors think the time should be kept in oder to make them consistent compared to the above figure.

38) **Comment:**P13/L9: ". . . quadrature detection. . ."both is true but stick to one term during the paper

   **Changes in Manuscript:** *quadrature demodulation* is replaced by *quadrature detection*

39) **Comment:**P13/L14: See also abstract. You are not consistent when you talk about the effective deadtime. In the abstract you refer to 4ms (3.6ms + 0.42ms? P13/L3+10) which is only the Rx filtering and here you include the artefact due to excitation current decay (5.8ms). Personally, I think that ApsuRx has an effective dead time of 4ms but dependent on the used Tx you should clip the data to 6ms to avoid pulse artifacts. Once Apsu includes a Tx you should provide the maybe longer effective deadtime for the whole SNMR system. Please consider to avoid calling it effective deadtime here and change the sentence to 5.8 ms including excitation current decay. . .

   **Changes in Manuscript:** *including excitation current decay* has been added following *5.8 ms.*

40) **Comment:**P14/L3: "...well-established surface-NMR Rx system. . . "

    **Changes in Manuscript:** *instrument* is changed to *surface-NMR Rx system.*

41) **Comment:**P14/L6: The Apsu receiver system might be misleading as you presented ApsuRx. Maybe introduce the System consisting of one Apsu Master and two ApsuRx first. E.g. "The used Apsu receiver system consists of one Apsu Master and two ApsuRx, One channel of an ApsuRx was . . .."

    **Changes in Manuscript:** *consists of one Apsu Master and two ApsuRx* is inserted after *Apsu receiver system.*

42) **Comment:** Figure 11: "after being scaled to the GMR signal by the area-turn factor of the coils (1200/800)"

    **Changes in Manuscript:** *to the GMR signal* is inserted after *after being scaled.*

43) **Comment:**P15/L7: "The signal recorded by the two Rx instruments (GMR, Apsu) were processed. . ."

    **Changes in Manuscript:***receivers* is replaced by *Rx instruments (GMR, Apsu).*

44) **Comment:**P16/8: ". . .receiver system where multiple Apsu Rx units each connecting . . . connected to an ApsuMaster"

    **Changes in Manuscript:** *We presented a new multichannel surface-NMR receiver system where ApsuRx units connecting up to three receiver coils are wirelessly connected to the ApsuMaster* is replaced by *We presented a new multichannel surface-NMR receiver system where multiple ApsuRx units each connecting up to three receiver coils are wirelessly connected to an ApsuMaster.*

45) **Comment:** P16/9: see previous comments on improving SNR. You do not compare the SNR properties of your system to another system. While Apsu might be a significant upgrade to your NUMIS, the in detail presented features to improve SNR (RNC, short dead time) are state-of-the-art (GMR, MIDI Radic). The impact of the new features (Wireless connection (outlook), dual gain recording, differential coils+ Rx) to improve SNR are not shown. Please simply rephrase it to ". . . the aim of the receiver system is a high SNR and . . ."

    **Changes in Manuscript:** *to improve SNR* is replaced by *to Rx loops deployability.*

46) **Comment:** P16/11+16ff: see previous comments on widely separated Rx and Ref loops. Please add a comment that modify this statement. While a long distance between Rx and Ref loops is technically possible with Apus, I have strong doubts that RNC will perform well or even improve

    **Changes in Manuscript:** Please refer to the response to the general comment 3.

**References**

Chen Chen, Fei Liu, Jun Lin, and Yanzhang Wang. Investigation and optimization of the performance of an air-coil sensor with a differential structure suited to helicopter TEM exploration. *Sensors*, 15(9):23325–23340, 2015.

Jakob Juul Larsen, Esben Dalgaard, and Esben Auken. Noise cancelling of MRS signals combining model-based removal of powerline harmonics and multichannel Wiener filtering. *Geophys. J. Int.*, 196(2):828–836, 2014.

Anatoly Legchenko, Jean-Michel Baltassat, Alain Beauce, and Jean Bernard. Nuclear magnetic resonance as a geophysical tool for hydrogeologists. *Journal of Applied Geophysics*, 50(1-2):21–46, 2002.

Mike Müller-Petke, Martina Braun, Marian Hertrich, Stephan Costabel, and Jan Walbrecker. Mrsmatlab - A software tool for processing, modeling, and inversion of magnetic resonance sounding data. *Geophysics*, 81(4): WB9–WB21, 2016.

Nicklas Skovgaard Nyboe and Kurt Sørensen. Noise reduction in TEM: Presenting a bandwidth-and sensitivity-optimized parallel recording setup and methods for adaptive synchronous detection. *Geophysics*, 77(3):E203–E212, 2012.

T Radic. Improving the signal-to-noise ratio of surface nmr data due to the remote reference technique. In *Near Surface 2006-12th EAGE European Meeting of Environmental and Engineering Geophysics*, 2006.

David O Walsh. Multi-channel surface NMR instrumentation and software for 1D/2D groundwater investigations. *J. Appl. Geophys.*, 66(3):140–150, 2008.

---

## Author Comment (AC2) · 4 Jul 2018

**Response to Referees' Comment on "Apsu: a wireless multichannel receiver system for surface-NMR groundwater investigations" by Lichao Liu et al.**

Lichao Liu et al.
lichao@geo.au.dk

To Referee Trevor Irons (gi-2018-1-RC2): Dear Referee, Thank you for reviewing our manuscript and raising issues that help to improve the manuscript. The authors response to all the comments in the following context and a marked-up manuscript is appended.

**1 Response to General comments**

1. **Comment 4:** Are the scientific methods and assumptions valid and clearly outlined? Yes, the system is described in a fair degree of detail. A better description or citation of differential coils should be included. I would also like to see a description of the power requirements of the receivers, and how long data collection can be performed on a single charge.

   **Response and change in manuscript:** Two references Nyboe and Sørensen [2012], Chen et al. [2015] addressed the differential Rx coil in airborne TEM are cited. Compared with the traditional coil, the differential coil has three output end: the positive, ground and negative.

   The power consumption of the receiver box is around 6.5 W (12 V, 0.54 A) due to the microprocessor is a low power-consumption FPGA chip. The power consumption will increase by approximately 5 W when the WiFi antenna is used. Also due to the efficiency of the power convert in the board, the 9 Ah lithium battery can support the whole system for 6 - 8 hours. Based on the field experiences, a single battery can be used for a day's measurement.

   **Change in manuscript:** The sentence *Differential Rx coil is beneficial to cancel the common-mode noise and it is able to reduce the Johnson noise of the coil by half [Nyboe and Sørensen, 2012, Chen et al., 2015]. The typical common mode noise is the induced noise in the leading cable and the wiring of the acquisition board by the coupling noise.* is added.

   A sentence describes the power consumption is added in the 3.1 WiFi network section: *The power consumption of a ApsuRx including the Wi-Fi antenna is around 12 W and 6 - 8 hours of data collection can be performed on a single charge of a 9 Ah battery. Batteries can be hot-swapped in the system without interrupting data collection.*

2. **Comment 5:** Are the results sufficient to support the interpretations and conclusions? The authors claim to have improved upon the SNR of the measurements. However, the field example does not demonstrate a reduced noise floor compared to other available instrumentation. If the authors should substantiate this statement, or remove it from the manuscript.

   **Response:** Agree, the main features of our wireless receiver are increased Rx loops deployability and reduced effort in field measurements. The collected data is normally dominated by the couping EM noise. The actual gain in SNR obtained with this new field strategy is heavily dependent on the site-specific noise conditions. But it is potential to improve SNR in some scenarios which will be demonstrated in detail later. Due to the practical reason, we cannot compare the developed receivers with the existing system directly.

3. **Comment 7:** Do the authors give proper credit to related work and clearly indicate their own new/original contribution? On page 2, line 10 a description of the first surface NMR instruments cites Legchenko and Valla, 2002 which describes the Iris NUMIS. However, the first surface NMR instrument was the Hydroscope:

   In this circumstance, it would be appropriate to cite the first instrument in addition to the NUMIS.

   **Change in manuscript:** The reference Semenov [1987] is added.

4. **Comment 8:** Does the title clearly reflect the contents of the paper? It does, however the acronym(?) 'Apsu 'is never defined. If it has some sort of meaning, please define it in the copy somewhere.

   **Response:** 'Apsu 'is just the name of our system not a acronym. You can also see this link
   https://en.wikipedia.org/wiki/Abzu.
   It was the name for fresh water from underground aquifers which was given a religious fertilising quality in Sumerian and Akkadian mythology.

5. **Comment 9:** Does the abstract provide a concise and complete summary? It does, however the discussion of SNR improvements either need to be substantiated or removed from the abstract as well.

   **Response:** Please refer to the second response.

6. **Comment 10:** Is the overall presentation well structured and clear? The paper is well structured, with the exception of §4.3 §4.4 and §5. The dead time discussion and (to some extent) field noise synthetics follows from the field examples. It would be more clear to introduce the field cites as e.g. 'Field Validations 'with subsections dedicated to dead time realizations and noise synthetics. As it stands Schillerslage is introduced twice and Silkeborg once. If the authors want to keep the Silkeborg examples in §4 that would be fine, but I would still recommend moving the dead time to §5 with a new §5.2 describing the data comparisons. If the phase is presented (discussed below), this could be a separate section as well.

   **Change in manuscript:** The dead time subsection in §4 is moved to the second subsection in §5.

7. **Comment 11:** Is the language fluent and precise? The manuscript is well written. On page 10 line 5 a trailing apostrophe (') is used where a leading apostrophe (') should be.

   **Change in manuscript:** The (') has been changed to (') on Page 12 Line 5.

8. **Comment 13:** Should any parts of the paper (text, formulae, figures, tables) be clarified, reduced, combined, or eliminated? See above discussion of §4 and §5.

   **Change in manuscript:** Please refer to the response to comment 10.

9. **Comment 14:** Are the number and quality of references appropriate? While the use of GPS timing is novel in surface NMR, it is common in the MT/CSEM community. For example the Zonge Zen system. A citation of this prior art would be appropriate and also affirm that GPS timing can reliably be used. Seismic nodal systems also use GPS timing and can be cited.

   **Change in manuscript:** A reference Sens-Schönfelder [2008] about GPS receiver in seismic is added.

**2  Response to suggestions**

1. **Comment 1:** The use of the word 'identical 'on P. 3 line 5 to describe the recorded NMR signals should be avoided. This description gives the impression that the two signals have no discernable measure between. 'Practically equivalent 'or some similar verbiage would be preferable.

   **Change in manuscript:** 'Practically equivalent 'replaces 'identical '.

2. **Comment 2:** The jet colourmap in figure 11 should be replaced with a perceptually uniform one. Additionally, the colormap clips at 0, but the quadrature detection should result in negative values as well. A diverging colormap centred around 0 is highly encouraged for the top two subfigures.

   **Response:** Due to the phase between two systems are not equal. Only the absolute amplitudes are shown in Fig. 11. Therefore, we can only find values larger than 0.

3. **Comment 3:** Complex inversion is an important consideration in surface NMR, especially with separated transmitters and receivers. Data phase comparisons (or real/imaginary plots) with the GMR are highly encouraged and will confirm that the developed instrumentation is at the bleeding edge of surface NMR instrumentation.

   **Response:** There exists an internal offset between the measured NMR signal by GMR and the true signal phase. To our knowledge the magnitude of the internal phase for the GMR is not known, or at a minimum is not commonly removed from data without using an inversion process. For this reason we base the comparison on amplitude data.

   Second, our receivers is not critically synchronous with the GMR system. Third, the sampling frequency in two systems are not the same. Hence, the measured phase by GMR and Apsu were not equal in that measurements because of differences in the internal phases that are present in the observed data.. But the measured phase by Apsu matches with the modeled phase from a well-know resistivity model by using our own new-built transmitter. A more rigorous accounting of internal phase corrections for the Apsu system is the subject of a parallel research paper. This non-trivial correction is heavily influenced by the transmitter (which is not discussed in this paper), therefore we prefer to base the validation on the amplitudes.

**References**

Chen Chen, Fei Liu, Jun Lin, and Yanzhang Wang. Investigation and optimization of the performance of an air-coil sensor with a differential structure suited to helicopter TEM exploration. *Sensors*, 15(9):23325–23340, 2015.

Nicklas Skovgaard Nyboe and Kurt Sørensen. Noise reduction in TEM: Presenting a bandwidth-and sensitivity-optimized parallel recording setup and methods for adaptive synchronous detection. *Geophysics*, 77(3):E203–E212, 2012.

A. G. Semenov. NMR hydroscope for water prospecting. In *Expanded Abstracts*, pages 66–67. Indian geophysical Union, 1987. Proceedings of the Seminar on Geotomography, Hyderabad.

Christoph Sens-Schönfelder. Synchronizing seismic networks with ambient noise. *Geophys. J. Int.*, 174(3):966–970, 2008.